# Endogenous Polyamines and Ethylene Biosynthesis in Relation to Germination of Osmoprimed *Brassica napus* Seeds under Salt Stress

**DOI:** 10.3390/ijms23010349

**Published:** 2021-12-29

**Authors:** Katarzyna Lechowska, Łukasz Wojtyla, Muriel Quinet, Szymon Kubala, Stanley Lutts, Małgorzata Garnczarska

**Affiliations:** 1Department of Plant Physiology, Institute of Experimental Biology, Faculty of Biology, Adam Mickiewicz University, Poznań, ul. Uniwersytetu Poznańskiego 6, 61-614 Poznań, Poland; katarzyna.lechowska@amu.edu.pl (K.L.); lukasz.wojtyla@amu.edu.pl (Ł.W.); 2Groupe de Recherche en Physiologie Végétale (GRPV), Earth and Life Institute–Agronomy (ELI-A), Université Catholique de Louvain, Croix du Sud 45, boîte L7.07.13, B-1348 Louvain-la-Neuve, Belgium; muriel.quinet@uclouvain.be (M.Q.); stanley.lutts@uclouvain.be (S.L.); 3Institute of Biochemistry and Biophysics, Polish Academy of Science, ul. Pawińskiego 5A, 02-106 Warszawa, Poland; szymon.globus@ibb.waw.pl

**Keywords:** germination, polyamines, priming, salinity, seeds

## Abstract

Currently, seed priming is reported as an efficient and low-cost approach to increase crop yield, which could not only promote seed germination and improve plant growth state but also increase abiotic stress tolerance. Salinity represents one of the most significant abiotic stresses that alters multiple processes in plants. The accumulation of polyamines (PAs) in response to salt stress is one of the most remarkable plant metabolic responses. This paper examined the effect of osmopriming on endogenous polyamine metabolism at the germination and early seedling development of *Brassica napus* in relation to salinity tolerance. Free, conjugated and bound polyamines were analyzed, and changes in their accumulation were discussed with literature data. The most remarkable differences between the corresponding osmoprimed and unprimed seeds were visible in the free (spermine) and conjugated (putrescine, spermidine) fractions. The arginine decarboxylase pathway seems to be responsible for the accumulation of PAs in primed seeds. The obvious impact of seed priming on tyramine accumulation was also demonstrated. Moreover, the level of ethylene increased considerably in seedlings issued from primed seeds exposed to salt stress. It can be concluded that the polyamines are involved in creating the beneficial effect of osmopriming on germination and early growth of *Brassica napus* seedlings under saline conditions through moderate changes in their biosynthesis and accumulation.

## 1. Introduction

Soil salinity is an increasingly severe problem worldwide, posing a severe threat to food security due to its negative impact on agricultural productivity and sustainability. The area under soil salinity is gradually spreading on all continents. According to data provided by the Food and Agriculture Organization via the Global Map of Salt-affected Soils (GSASmap) tool, more than 3% of global topsoils and more than 6% of global subsoils are affected by salinity or sodicity [1]. Soil salinity affects more than 100 countries, and salt-affected soils occupy over 20% of the global irrigated area, increasing the range each day [2,3]. Salinity is one of the most significant abiotic stresses that impairs plant growth, development, and yield. Only 2% of terrestrial plant species are salt tolerant (halophyte). Most plants, including almost all crops, belong to glycophytes, displaying a relatively low salt tolerance [4]. However, the degree of salt tolerance varies among plant species and even cultivars [5]. 

An excess of soluble salt induces a double constraint on plants: first, a decrease in external water potential compromises water uptake (osmotic component), and second, the accumulation of toxic ions induces nutritional disorders [6]. Accordingly, the damaging effect of salinity on plants is a multifaceted issue. Depending on factors such as exposure time and severity, salt stress affects plants in several ways; it induces or is accompanied by water-, osmotic-, oxidative-stress and nutritional imbalance, ultimately leading to various physiological and metabolic disorders [5,7,8]. Plant response to stress depends on the developmental stage [9]. Seed germination and seedling establishment are the most critical phases in the plant life cycle that encounter constantly changing environmental conditions, including salt stress [10,11]. Salinity reduces the rate and percentage of germination and negatively affects all key events occurring during germination from imbibition, through metabolism activation, elongation of embryonic tissue, to seedling establishment [12]. Therefore, efforts to improve stress tolerance during germination appear to be important to allow successful seedling emergence, plant growth and maintenance of crop productivity under salinity conditions. In this regard, seed priming is a promising direction in modern agriculture as a relatively low-cost and straightforward method with a well-documented positive impact on seed germination and further plant growth, particularly under unfavorable environmental conditions [13].

Seed priming is a presowing seed treatment that relies on controlled, partial seed hydration, leading to the activation of early germination events, however, insufficient for radicle protrusion through the seed coat. Since the completion of germination does not take place, seeds are still desiccation tolerant and can be dried back to their original moisture without loss of viability. Depending on the chemical nature of the priming agent, several seed priming methods are distinguished, including hydropriming, osmopriming, hormone priming, and others [14,15]. Osmopriming refers to seed soaking in an osmotic solution with low water potential such as polyethylene glycol (PEG) and belongs to the most commonly used types of priming treatment. Seed priming with PEG has been shown as an effective method to improve seed germination, seedling emergence, and stress tolerance of several crop plants under unfavorable conditions such as salt, water, chilling, and nano-ZnO stresses [14]. The priming procedure itself may generate a moderate abiotic stress during both soaking (e.g., osmotic stress and drought created by the priming agents) and dehydration steps. It is tempting to propose that priming cannot simply be considered as an acceleration of germination-related processes but also involves other specific mechanisms that improve germination and allow the seeds to cope with environmental stresses during seedling establishment [16].

Numerous studies showed multiple benefits of seed priming, including improved rate, percentage, germination uniformity, and seedling ability to emerge under salinity conditions. The alleviation of salt stress effects and increased tolerance induced by a priming treatment were reported in several crops, such as wheat (*Triticum aestivum* L.) [17,18] rice (*Oryza sativa* L.) [19], maize (*Zea mays* L.) [20,21], soybean (*Glycine max* L.) [22], sorghum (*Sorghum bicolor* L.) [23] and others [24]. Our previous studies demonstrated a positive impact of osmopriming on rape (*Brassica napus* L.) seed germination in optimal conditions [16,25] as well as under salt stress [26]. Rape is cultivated mainly for its oil-rich seed. In 2020–2021, world production of rape was estimated at 75.0 million tonnes, ranking second worldwide just after soybean [27]. Plants adapt to unfavorable environmental conditions by accumulating low molecular-weight compounds, such as proline (Pro) and polyamines (PAs). Among others, polyamines are involved in ion stress response. Salinity, heavy metals, iron excess or aluminum are toxicities that were reported to induce modifications in polyamine metabolism [28].

Polyamines are organic cations found in various organisms, performing diverse biological functions [29]. Putrescine (Put), spermidine (Spd), and spermine (Spm) are the main PAs in plants, and they are involved in the regulation of diverse physiological processes and responses to biotic and abiotic stresses. However, it is still not fully understood how PAs regulate plant growth and stress responses [30]. Advances in polyamine research in relation to environmental constraints considerably benefit from controlling the conversion between free, conjugated and bound forms. Due to their positive charge, PAs can bind various high molecular weight compounds by hydrogen and ionic binding, electrostatic and hydrophobic interactions, including DNA, RNA, chromatin and proteins, which can cause stabilization or destabilization of these large molecules. Furthermore, they can covalently conjugate to endoglutamines of proteins by the action of transglutaminase. The balance of free and conjugated PAs is critical for different developmental processes and the relative proportions of each form may vary among different species [28]. There are still many questions to answer regarding the roles of PAs and their free, conjugated and bound forms in regulating the response to salinity stress. However, it is presumed that the ratio between polyamines is more important than the concentration of individual forms. 

The diamine putrescine is produced either from ornithine or from arginine through the action of ornithine decarboxylase (ODC; EC 4.1.1.17) or arginine decarboxylase (ADC; EC 4.1.1.19), respectively. Spermidine and spermine are then synthesized from putrescine through the addition of aminopropyl groups transferred from decarboxylated S-adenosylmethionine (dcSAM) in reactions catalyzed by spermidine synthase (EC 2.5.1.16) and spermine synthase (EC 2.5.1.22). DcSAM is produced from S-adenosylmethionine (SAM) by S-adenosylmethionine decarboxylase (EC 4.1.1.50) [28]. SAM is also the precursor of ethylene: it can be metabolized to 1-amino cyclopropane carboxylic acid and then ethylene. Ethylene is a well-known senescing and stress-related gaseous hormone, but it also plays a crucial role in the regulation of germination [31]. Besides the classical aliphatic amine, which behaves as polycation under cellular pH conditions, an aromatic amine tyramine (Tyr) is produced from tyrosine decarboxylation and was reported to have some roles in plant response to abiotic stress [32,33].

This work aimed to better understand the beneficial effect of osmopriming on seed germination by estimating the modulation of the endogenous polyamine accumulation pattern during rape seed germination under salt stress. To gain more insight into PA metabolism, ODC and ADC activities were determined. As changes in PA metabolism may affect the production of ethylene, ethylene emission was also analyzed. This paper extends our previous paper related to osmopriming improved germination of *Brassica napus* seeds under salinity stress [26].

## 2. Results

### 2.1. Priming Improves Seed Germination under Salt Stress

In the experimental layout used in this paper, the primed and unprimed seeds germinating on water or NaCl were collected at the same stage during germination. This experimental design gives a chance to identify changes linked to priming treatment but not directly connected with the more advanced development of germinating primed seeds. Thus, to perform analyses on primed and unprimed seeds at the same developmental stages, unprimed seeds were collected after 11 h of germination in H_2_O (UP_11H___2_O_) and 16 h in 100 mM NaCl (UP_16NaCl_), which corresponds to reaching 1% of germinated seeds under the conditions mentioned above. Early visible radicle protrusion of primed seeds germinating under both salinity and control conditions occurred after 7 h of imbibition (P_7NaCl_, P_7H_2_O_, respectively) (Figure 1). The primed dried seeds (P_d_), i.e., seeds soaked in priming agent (PEG) and dried to initial moisture content, and unprimed dry seeds (UP_d_) were used as a control.

### 2.2. Seed Priming Modulates Pattern of Endogenous Polyamine Accumulation during Germination under Salt Stress

To investigate whether osmopriming confers tolerance to salt stress through the action of polyamines, we used the high-performance liquid chromatography (HPLC) method for the quantification of putrescine, spermidine, and spermine in free, conjugated, and bound forms during germination of primed and unprimed *Brassica napus* seeds exposed to salt stress.

#### 2.2.1. Seed Priming Modulates the Content of Free Polyamines

##### Free Putrescine

During germination of primed seeds under both optimal conditions and in the presence of NaCl, no changes in the content of free putrescine were observed as compared to primed dried seeds—its level remained the same in all tested samples (P_d_, P_7H___2_O_, P_7NaCl_) (Figure 2). In unprimed seeds, the accumulation of free putrescine was unchanged during germination on the water compared to dry seeds. In contrast, under salt stress, a statistically significant increase of free putrescine by 46% was observed (UP_11H___2_O_ vs. UP_16NaCl_). Comparing seeds that underwent priming treatment with untreated ones, osmopriming led to a rise of the free putrescine level during germination in optimal conditions (by 30%) but had no impact on seeds exposed to salt stress.

##### Free Spermidine and Spermine

The changes of both free spermidine and spermine levels followed a broadly similar trend during seed germination under optimal conditions and during exposure to salt stress (Figure 2B,C).

The salt stress did not differentiate the level of free spermidine and spermine in primed seeds—the contents of these polyamines remained at the same level as in the case of primed seeds germinating under optimal conditions as well as in dry seeds. A different tendency was noticed in unprimed seeds, in which exposure to the stress caused a decrease in the level of free spermidine and spermine. Unprimed seeds germinating under salinity conditions (UP_16NaCl_) were characterized by statistically significantly lower free spermidine and spermine content than unprimed seeds germinating on water (UP_11H___2_O_) and seeds in the dry state.

Osmopriming treatment decreased the content of free spermidine and spermine in dry seeds but had no further impact on their level during postpriming germination under optimal conditions and in the presence of salt stress. The same contents of free polyamines characterized the osmoprimed seeds germinating in the presence of water (P_7H_2_O_) as unprimed seeds (UP_11H___2_O_). Under salt stress, there was a noticeable difference in the free spermidine and spermine accumulation between primed and unprimed seeds (Figure 2B,C). For osmoprimed seeds germinating 7 h on NaCl, the free spermidine and spermine level was 24% and 35% higher, respectively, than in unprimed seeds germinating under similar stress conditions (UP_16NaCl_).

#### 2.2.2. Seed Priming Modulates the Content of Conjugated Polyamines

##### Conjugated Putrescine

For primed seeds, the accumulation of conjugated putrescine was the highest in dry seeds and noticeably dropped during post-priming germination (Figure 2D). It was observed that salt stress did not affect the pool of conjugated putrescine in osmoprimed seeds since there was no difference in the content of conjugated putrescine between seeds germinating for 7 h in water and in the presence of NaCl. In turn, the putrescine level in unprimed seeds dropped during germination in the water while its maximum content was noted in seeds germinating for 16 h in the presence of NaCl—the accumulation of conjugated putrescine in UP_16NaCl_ was nearly twice as high as in UP_11H___2_O_.

In response to osmopriming, the level of conjugated putrescine was modulated in all tested conditions—evident differences between unprimed and primed seeds were observed in the dry state, during germination on the water as well as during germination in the presence of NaCl. Considering dry seeds and seeds germinating in optimal conditions, osmopriming led to an increase of conjugated putrescine compared to unprimed ones, while under stress conditions, the opposite trend was observed—the level of conjugated putrescine in osmoprimed seeds was nearly half that in unprimed ones.

##### Conjugated Spermidine

Our results showed no difference between the level of conjugated spermidine in primed and unprimed seeds in the dry state (Figure 2E). During germination in optimal conditions, unprimed seeds were characterized by a reduced content of conjugated spermidine compared to the UP_d_ seeds. Salt stress did not cause changes in the level of conjugated spermidine. In primed seeds germinating in the presence of water or NaCl, the level of conjugated spermidine dropped by about a quarter in relation to P_d_ seeds. It was noted that in primed seeds germinating on the water as well as under salt stress, the content of conjugated spermidine was lower than in unprimed ones germinating under the same conditions.

##### Conjugated Spermine

The highest concentration of conjugated spermine was noticed in dry seeds without differences in its content between UP_d_ seeds and P_d_ seeds (Figure 2F). During germination on water, conjugated spermine dropped to almost the same level in both unprimed and primed seeds. Exposure to stressful conditions did not significantly affect the accumulation of this polyamine: there was no statistically significant difference between P_7H___2_O_ seeds and P_7NaCl_ seeds nor between UP_11H_2___O_ and UP_16NaCl_ seeds. Osmopriming had no impact on conjugated spermine level during germination since there was no difference between seeds germinating in the presence of water (UP_11H___2_O_ vs. P_7H_2_O_) nor in the presence of NaCl (UP_16 NaCl_ vs. P_7NaCl_).

#### 2.2.3. Seed Priming Modulates the Content of Bound Polyamines

No modification of bound putrescine (Figure 2G) was recorded in primed and unprimed seeds in the dry state and during germination in optimal conditions. Salt stress caused a decrease of bound putrescine in both osmoprimed and unprimed seeds as compared to dry seeds and those germinating under optimal conditions, however, again, the content of bound putrescine between osmoprimed and unprimed seeds did not differ.

The pattern of changes in bound spermidine (Figure 2H) and bound spermine (Figure 2I) was almost identical to the pattern observed for bound putrescine. There was no difference in the content of bound spermidine between primed and unprimed seeds in the dry state and during germination in optimal conditions. However, both primed and unprimed seeds showed a decrease of bound spermidine during germination under salt stress as compared to germination on water. There were no differences in the concentration of bound spermidine between unprimed and primed seed germinating in the presence of NaCl. 

Concerning bound spermine (Figure 2I), the highest accumulation in primed seeds was observed during germination on water. Salt stress caused the reduction of bound spermidine to the initial level noticed in primed dried seeds. For unprimed seeds, there was no change in the level of bound spermidine between dry seeds and the seeds germinating on water. An apparent decrease in bound spermine concentration was observed in unprimed seeds exposed to salt stress. As with bound putrescine and spermidine, no differences were found in bound spermine concentration between unprimed and primed seeds (i.e., P_d_ vs. UP_d_, P_7H___2_O_ vs. UP_11H___2_O_, and P_7NaCl_ vs. UP_16NaCl_).

### 2.3. Seed Priming Modulates PAs Ratios

Table 1 presents the values calculated for the (Spd + Spm)/Put ratio and Put/Spd ratio. In this study, we considered the PAs ratios for all polyamine forms (sum of free, conjugated, and bound forms). In general, salt stress did not change (Spd + Spm)/Put ratio for primed seeds as compared to seeds germinating under optimal conditions, while the exposure of unprimed seeds to stress led to a reduction of the (Spd + Spm)/Put ratio. Moreover, under salinity conditions, priming enabled seeds to maintain a higher PA ratio compared to unprimed seeds. In the case of the Put/Spd ratio, salt stress did not affect the value of this parameter in primed seeds, while it caused a significant increase in unprimed ones.

### 2.4. Seed Priming Modulates ODC and ADC Activity during Seed Germination under Salt Stress

The effect of priming treatment on ODC and ADC is presented in Figure 3. Concerning ODC, priming treatment reduced its activity in dry seeds. During germination on water, ODC activity increased in both primed and unprimed seeds, reaching the same level. In turn, salt stress changed ODC activity in primed seeds, causing its further increase, while in unprimed seeds, ODC activity remained at the level observed for seeds germinating on water. The activity of ADC differed significantly between primed and unprimed seeds in all tested conditions. ADC activity increased in response to the priming treatment and achieved the highest value in primed seeds germinating under optimal conditions. Under salt stress, ADC activity significantly decreased in primed seeds, remaining at a higher level than in unprimed seeds germinating under the same stress conditions.

### 2.5. Seed Priming Modulates the Level of Free Tyramine

Unprimed seeds germinating under optimal conditions did not differ in tyramine level from unprimed seeds in a dry state but exhibited a lower level than unprimed seeds exposed to salt stress (Figure 4). The accumulation of tyramine in primed seeds increased during germination on water and slightly dropped to the initial level in response to stress conditions. Priming induced a significant increase of free tyramine in seeds when comparing primed and unprimed seeds. Free tyramine was nearly twice as high in primed dried seeds as in unprimed seeds in the dry state. Notably, the level of free tyramine was also higher in primed seeds germinating both in the presence of water and NaCl (by 72% and 44%, respectively) compared to unprimed ones.

### 2.6. Priming-Induced Changes in Ethylene Production during Early Seedling Growth under Salt Stress

Our results did not show any obvious differences in ethylene production during the first hours of germination between primed and unprimed seeds, both germinating in the presence of water or NaCl (Figure 5). However, significant differences were noticed later during early seedling growth. Ethylene production in the tested samples resembles exponential growth. For primed seeds, ethylene emission increased after 21 h of germination under optimal conditions and after 26 h under salt stress. For unprimed seeds, the low and relatively stable level of ethylene production was maintained up to 43 h from the beginning of germination for seeds germinating in the presence of water and 46 hours for seeds germinating in the presence of NaCl. Salt stress slowed down ethylene production as compared to control conditions in both primed and unprimed seeds, but it should be noted that the level of ethylene was about 10 times higher for seedlings issued from primed seeds and exposed to salt stress compared to seedlings grown from unprimed seeds.

## 3. Discussion

Polyamines are involved in a myriad of processes in plants, including germination. Seed germination is an extremely dynamic process that requires metabolism activation, biogenesis of organellar structure, mobilization of stored substances, cell elongation, replication, and cell division. Literature data indicate the role of polyamines in a wide range of biological processes during plant development, from germination to flowering [30]. In most of the studied plant species, the increase in polyamine content was noted during the early stages of germination [30]. PAs are also engaged in seed dormancy release, although there is some variation in the fluctuation of PA content during germination, and the type of PAs, mostly active as enhancers of germination, varied depending on the plant species [34]. 

The impact of seed priming on the polyamine metabolism has not been studied in detail, so far. Our results enhance the hypothesis about the function of polyamines in the acquisition of stress tolerance by plants, which, from an agrotechnical point of view, is extremely important at germination and early seedling development. Basra et al. [35] reported that osmopriming of onion seeds enhanced the accumulation of Put and Spd, while the level of Spm accumulation remained at a similar level to that of unprimed seeds. In *Brassica napus,* we noticed an increase only in Put accumulation in seeds after priming. However, priming influences the ratio of the free, bound, and conjugated fraction of polyamines, which may affect their metabolic activity [30]. It has been demonstrated in numerous experimental systems that the ratio between polyamines is a more important regulating factor in plant tissues than the absolute concentration of individual polyamines [28]. 

Put accumulation occurs in salt-stressed plants, particularly in halophytes [36]. In this study, the total amount (Put + Spd + Spm) of free PAs was higher in germinating primed seeds both under control and stress conditions than in unprimed seeds. In contrast, the total amount of conjugated PAs was higher in unprimed seeds. Spd content (free as well as other forms) was higher throughout the germination process (imbibition and metabolism activation phases) in both primed and unprimed variants as compared to Put and Spm (Appendix A). These results are in line with those observed in *Arabidopsis thaliana*, *Malus domestica*, *Cedrela fissilis*, and sweet corn, where Spd content was also higher than other PAs [37,38,39,40]. In *Arabidopsis*, the addition of exogenous Put (but not Spd or Spm) resulted in the inhibition of germination [41], which suggests the function of Put in the regulation of germination. Exogenous treatment with Spm or Spd improved soybean seedlings’ root and shoot growth under excess moisture stress by altering the enzymatic antioxidant system [42]. The protecting role of PAs on membranes, nucleic acids, proteins, and biomolecules has been well established [33]. Since seed priming also results in reduced lipid peroxidation [35,43,44,45], polyamines may also act to protect the membranes against oxidative damage due to their radical-scavenging properties and ability to interact with phospholipids [46]. One of the biochemical markers of seed germination could be the PA ratio [(Spd + Spm)/Put] which increased during seed imbibition, pointing to the importance of Spm and Spd in the period prior to radicle protrusion [47]. It was suggested that the increase in Spm and Spd levels is essential during cell elongation, a crucial event in the germination process [47]. In our study, the highest PA ratio was observed in unprimed dry seeds (UP_d_) as well as in unprimed seeds germinated on water (UP_11H___2_O_), whereas the lowest PA ratio was observed in unprimed seeds germinated under salt stress (Table 1). Paul and Roychoudhury [48] showed that salinity increased (Spm + Spd)/Put ratio more in the tolerant cultivar, thus, a higher level of (Spm + Spd)/Put ratio in primed versus unprimed seeds germinating under salt stress conditions could be the premise to consider primed seeds as more tolerant to salt stress than unprimed seeds. Additionally, it seems to be confirmed in the germination test and our previously published results [26]. It could be assumed that the activation of the pre-germinative metabolism during osmopriming also includes changes in the PA ratio. The Put/Spd ratio was also proposed as a good indicator of stress in plants [49], which was higher in primed seeds than in unprimed ones when they germinated on water. In contrast, the Put/Spd ratio increased in unprimed seeds about 60% under salt stress but remained unchanged in primed ones (Table 1). A higher increase in the Put/PAs ratio under NaCl stress was observed in salt-sensitive cultivars for several species [50]. Such observation may suggest that a more stable PA ratio in primed seeds and seedlings grown from primed seeds helps maintain plant growth and development under salt stress conditions at the proper level. The higher Put/Spd ratio during postpriming germination on water seems to confirm the hypothesis, illustrating the physiology of priming-induced stress tolerance proposed by Chen and Arora [51]. The authors suggested that priming-induced stress tolerance could be achieved via osmopriming-related events that facilitate the transition of quiescent dry seed into a germinating state and lead to improved seed germination. Moreover, the imposition of abiotic stress on seeds during osmopriming represses radicle protrusion but stimulates stress response, potentially inducing cross-tolerance. The authors have suggested that these two strategies together constitute a “priming memory” in seeds, which can be recruited upon a subsequent stress exposure and manifested by higher stress tolerance of germinating osmoprimed seeds and developing seedlings. The increased Put/Spd ratio in seeds that underwent a priming procedure as well as a higher value of this ratio in primed seeds germinating on water than in unprimed seeds, speak in favor of this hypothesis and are in close agreement with our previous findings [16,25,26]. 

Modulation of the endogenous PA levels in plants observed under a wide range of abiotic stresses points to their important role in protecting plants against harsh environmental conditions [29]. Our results showed an elevation of the endogenous Put in primed seeds and in unprimed seeds germinated under salt stress. In primed dried seeds (P_d_), the increase in Put level was accompanied by the upregulation of ADC activity, whereas in unprimed seeds germinated under salt stress, it was with higher ODC activity (Figure 3). This result suggests that Put synthesis in unprimed salt-stressed seedlings of rape occurs mainly through ODC and not the ADC pathway. ODC was also more effectively upregulated in response to salt stress than ADC in Zoysiagrass [52]. The ADC pathway seems to be stimulated through the priming procedure, as ADC activity was upregulated in primed seeds both in a dry state and during germination on water, whereas in unprimed seeds, no changes in ADC activity were noticed. ADC was considered as the main enzyme involved in response to abiotic stresses [53]. Seed priming, as was discussed above, reflects stress exposure, what may contribute to increasing ADC activity in primed seeds. ADC is also essential for seed development in *Arabidopsis* [54]. The enzyme activity was enhanced during the imbibition of wheat seeds [55], which revealed its function in regulating PA level during germination. Moreover, it could be considered that a rapid seed imbibition may cause severe damage to plasma membranes, which may also trigger stress responses [56]. It was suggested that salt resistance was associated with an ability to increase Put synthesis as a consequence of both higher ADC and ODC activities [57]. Thus, priming-induced ADC activity could be a part of the complex mechanisms enhancing germination efficiency and stress tolerance. It should be noted that the actual level of PAs is a consequence of both synthesis and catabolism through amine oxidase, which was not analyzed in this study. 

PA biosynthesis is affected by different molecules, which can enhance or inhibit this process. One of them is ethylene. We observed an earlier and higher production of ethylene in germinating primed seeds which could be due to activation of metabolic processes in primed seeds as well as faster and more efficient germination. Ethylene was indeed proposed to play a key function in regulating seed dormancy and germination [58]. The presence of salt causes a decrease in ethylene production both in primed and unprimed seeds during germination (Figure 5), which seems to be coherent with the findings that ABA, which is one of the factors engaged in salt stress signalization and response, as well as regulation of germination under salt stress [59], inhibits ethylene synthesis through 1-aminocyclopropane-1-carboxylic acid synthase and 1-aminocyclopropane-1-carboxylic acid oxidase activities [58]. It was also stated that ethylene treatment could recover the germination rate of alfalfa seeds under salt stress [60]. The increase in ethylene production during germination under saline conditions has been reported in different species such as rice, wheat, maize, soybean and lettuce [61,62,63,64,65]. In our study, the lower evolution of ethylene was noticed under salt stress. A similar observation was also done during germination of *Stylosanthes humilis* seeds under salt stress, which was associated with enhanced ABA production [66]. 

The interaction between polyamines and signaling molecules, such as H_2_O_2_, is frequently studied in relation to plant stress physiology. Polyamines are linked to ROS metabolism through H_2_O_2_ production via their catabolism pathway. Exogenously applied Spd effectively alleviated the damage caused by salt stress by upregulating the antioxidant enzyme activity and expression of their genes in *Gladiolus gandavensis* seedlings [67]. Exogenous Spm treatment under salt stress influenced ROS metabolism by upregulating the gene expression of essential antioxidant enzymes superoxide dismutase and catalase in tomato and Mung bean [68]. Furthermore, the application of Put enhanced antioxidant enzymes activity under salt stress in Indian mustard [69]. In the present study, higher levels of free Put accumulation was noticed in primed seeds under both stress and control conditions and in unprimed seeds under salt stress (Figure 2A). The protective effect of Put under stress conditions due to the maintenance of redox homeostasis through the H_2_O_2_ signaling pathway was proposed for several biotic and abiotic stresses [30,33,70]. The accumulation of hydrogen peroxide was observed during the germination of primed and unprimed seeds under both control and stress conditions [26]. The role of ROS in signaling networks during germination was reviewed by Wojtyla et al. [71] and Bailly [72]. Hydrogen peroxide plays a dualistic function in living organisms, and it is also discussed as a positive regulator of germination, which may promote this process. In the study by Kubala et al. [26], an increased level of H_2_O_2_ was associated with Pro accumulation during the germination of primed rape seeds under salt stress conditions, which also corresponded to enhanced expression and enzymatic activity of pyrroline-5-carboxylate synthetase, a key enzyme involved in the Pro biosynthesis pathway. The correlation between Put and Pro level was stated by Ebeed et al. [73] and Paul et al. [74], who demonstrated that Pro accumulation occurs through the signaling pathway by PAs, which leads to the activation of pyrroline-5-carboxylate synthetase (*P5CS*) gene expression and downregulation of proline dehydrogenase (*PDH*) gene expression involved in Pro degradation. 

In the present research a similar trend was also noticed for tyramine (Figure 4), which is an aromatic monoamine and was reported to increase in response to NaCl and Cd stress [32]. Tyramine accumulation may also trigger the synthesis of proline [75]. Aziz et al. [75] reported that changes in tyramine accumulation reflected the level of proline accumulation. The level of proline in germinating primed and unprimed seeds under control and salt stress conditions observed in our previous study [26], seems to correlate with the tyramine level in this study. Modulation of tyramine level and its conjugation was also reported in response to the pathogen [76], wounding [77] and UV-C [78]. Conjugated tyramine is involved in cell wall fortification and may directly regulate lignification and suberization processes [77,78]. Conjugated tyramine was not considered in the study; however, we demonstrated the obvious impact of seed priming on this amine. 

The protective function of polyamines on plants under adverse environments, particularly under drought and salt stress, is widely accepted and well documented in studies with exogenously applied polyamines in a wide range of concentrations and on many plant species. Recent studies highlight the function of exogenously delivered polyamines in stress alleviation by regulation of protein production at both transcriptional and translational levels [68,79,80]. Our previous parallel transcriptomic and proteomic study revealed the main processes affected during the priming and postpriming germination of rape seeds, indicating the important role of processes associated with protein metabolism: synthesis, post-translational processing, targeted proteolysis, and their turnover [26]. Present observations seem to confirm the previous statement that presowing seed priming is manifested during salt stress exposure rather through moderate than dynamic modulation of metabolism when compared to unprimed seeds [26]. This seems to speak in favor of the hypothesis of stress imprint as a consequence of priming. As regards a comparison of changes between polyamine fractions (free vs. conjugated vs. bound), the greatest differences between the corresponding pairs of primed and unprimed seeds were visible in the free (Spm) and conjugated (Put, Spd) fractions (Figure 2). It can be concluded that the involvement of polyamines in creating the beneficial effect of osmopriming on germination and early growth of *Brassica napus* seedlings observed under saline conditions takes place through changes in these two fractions, i.e., free and conjugated because in the case of bound polyamines there were no differences between the corresponding variants of primed and unprimed seeds (Figure 2). This could be another clue answering the question of why primed seeds have improved tolerance to stress factors, however, the detailed role of polyamines in stimulating germination and stress tolerance recruited through priming should be recognized in future studies.

## 4. Materials and Methods

### 4.1. Seed Osmopriming Procedure

The seeds of *Brassica napus* L. cv Libomir (kindly provided by OBROL Kulczyński Sp.J., Kruszewnia, Poland) were surface sterilized and primed in PEG 6000 solution (osmotic potential −1.2 MPa) for 7 days at 25 °C in the darkness on Petri dishes lined with 3 layers of filter paper wetted with PEG, according to the procedure described in our previous studies [16,25,26]. After soaking, seeds were washed 3 times with sterile deionized water to remove the osmotic agent and dried for 48 h at room temperature until they reach the initial moisture content (water content 5%; primed dried seeds, P_d_). This priming protocol was the most effective in terms of improved germination. Primed dried seeds (P_d_) were collected for germination or frozen in liquid nitrogen and stored at −80 °C.

### 4.2. Germination Conditions, Gemination Test and Experimental Design

The primed dried (P_d_) and unprimed dry (UP_d_) seeds were placed in plastic Petri dishes (1 g of seeds per Petri dish, ~200 seeds) on the top of the three layers of filter paper wetted with 10 mL of deionized water or 100 mM NaCl and germinated in darkness at 25 °C. Primed seeds germinating for 7 h in H_2_O (P_7H___2_O_, or NaCl (P_7NaCl_) were collected and frozen in liquid nitrogen and stored at −80 °C for future use.

The imbibition time of 7 h corresponds to the achievement of 1% germinated primed seed under both salinity and control conditions, representing end of seed germination sensu stricto (time point just prior to radicle emergence) [26]. To perform analyses on primed and unprimed seeds at the same stage during germination, unprimed seed were collected (without rinsing) after 11 h of germination in H_2_O (UP_11H___2_O_) and 16 h in 100 mM NaCl (UP_16NaCl_), which corresponded to reaching 1% of germinated seeds under the conditions mentioned above [26]. To analyze the pattern of endogenous polyamines and the activity of polyamines metabolizing enzymes, the following seed samples were used: P_d_, U_Pd_, P_7H___2_O_, UP_11__H___2_O_, P7_NaCl,_ and UP_16NaCl_. The experimental layout of ethylene measurement has been further expanded by unprimed and primed seeds germinating for 24 h in 10 mL of H_2_O and 10 mL of 100 mM NaCl.

### 4.3. Extraction and Quantification of PAs and Tyramine

Extraction of free, conjugated, bound polyamines (PAs) and tyramine was conducted according to the method described by Quinet et al. [57] with slight modifications.

#### 4.3.1. Free Polyamines and Tyramine Extraction

Seed samples frozen in liquid nitrogen (500 mg FW) were ground to a fine powder with 500 µL 4% ice-cold HClO_4_, vortexed vigorously, and then left to stand for 1 h in an ice bath, mixed regularly. The extract was centrifuged at 23,100× *g* for 20 min at 4 °C. The supernatant was used for dansylation or the extraction of conjugated PAs. The pellet was used to extract bound PAs.

#### 4.3.2. Conjugated Polyamine Extraction

The supernatant obtained from the free PAs extraction (200 mL) was mixed with 200 mL of 12 N HCl and vortexed for a few seconds at a slow speed. To perform acid hydrolysis, samples were heated at 110 °C for 16 h in hermetically closed glass bottles. HCl was then evaporated from the bottles at 80 °C, and the residue was suspended in 200 μL of 10% PCA and used for dansylation.

#### 4.3.3. Bound Polyamine Extraction

The pellet, which remained after the extraction of free PAs, was washed by adding 500 µL 4% HClO_4_, vortexed vigorously, and centrifugated at 23,100× *g* for 20 min at 4 °C. The supernatant was discarded, and the procedure was repeated a second time. After washing, the pellet was suspended in 500 µL of 1 N NaOH, vortexed vigorously, and left to stand for 30 min in an ice bath, mixed regularly. The extract was centrifuged at 23,100× *g* for 20 min at 4 °C. The collected supernatant was hydrolyzed according to the procedure described in the section on conjugated polyamine extraction.

#### 4.3.4. Sample Dansylation and HPLC Analysis

Polyamine extracts were derivatized with dansyl chloride as described by Lefèvre et al. [81] and analyzed by HPLC under conditions described by Quinet et al. [82]. Samples were re-suspended in methanol and filtered (Chromafil PES-45/15, 0.45 m; Macherey-Nagel, Duren, Germany). HPLC analysis was performed with Nucleodur C18 Pyramidcolumn (125 × 4.6 mm internal diameter, 5 µm particle size; Macherey-Nagel GmbH & Co. KG, Dueren, Germany) on a Shimadzu HPLC system coupled to an RF-20A fluorescence detector (Shimadzu, ‘s-Hertogenbosch, The Netherlands) at 340 nm and emission at 510 nm.

### 4.4. Enzyme Assay 

Extraction and assay of ADC and ODC enzymes were performed according to Quinet et al. [57]. Enzyme activity was determined by detecting the release of (C14) CO_2_ using (C14) labeled substrates (measuring CO_2_ evolution from the decarboxylation reaction). Protein content was measured by the protein-dye binding method according to Bradford [83], using bovine serum albumin as standard.

### 4.5. Ethylene Measurement 

The ethylene production was measured using ethylene detector ETD-300 (Sensor Sense, Nijmegen, The Netherlands) according to Critescu et al. [84]. Seeds sample (1 g) was placed in Petri dishes lined with three layers of filter paper moistened with 10 mL of deionized water or 100 mM NaCl. As a control from the obtained emission rates, the level of ethylene was measured in a similar cuvette without seeds. The measurements were carried out in the darkness, in stop-and-flow mode, with each cuvette being alternatively flushed with a flow of 3 L h^−1^ for 15 min for 71 h.

### 4.6. Statistical Analysis

All analyses were conducted at least in three independent experiments (three biological and two technical replicates). Germination tests were carried out on ten replicates of 100 seeds. The statistical deviation of the mean values was calculated using one-way analysis of variance (ANOVA) and the Tukey–Kramer multiple comparison test. The means were considered as significantly different at *p* < 0.05. Results that showed no statistically significant differences are marked with the same letter.

## 5. Conclusions

The present results shed a new light on the mechanisms underlying the osmopriming-induced improvement of seed germination and salt stress tolerance. The above findings are summarized in Figure 6. Priming-induced modulation of metabolites and enzyme activities is more evident in the germinating osmoprimed seeds in relation to the corresponding unprimed variations than between osmoprimed seeds germinating under stress and unstressed conditions. Then, it could be concluded that presowing seed osmopriming conveys an enhanced ability to respond to future events by modulating biochemical pathways, including those associated with polyamine metabolism. Overall, it may be assumed that the differences between stressed and unstressed variants are more evident rather in germinating unprimed seeds than in primed ones.

## Figures and Tables

**Figure 1 ijms-23-00349-f001:**
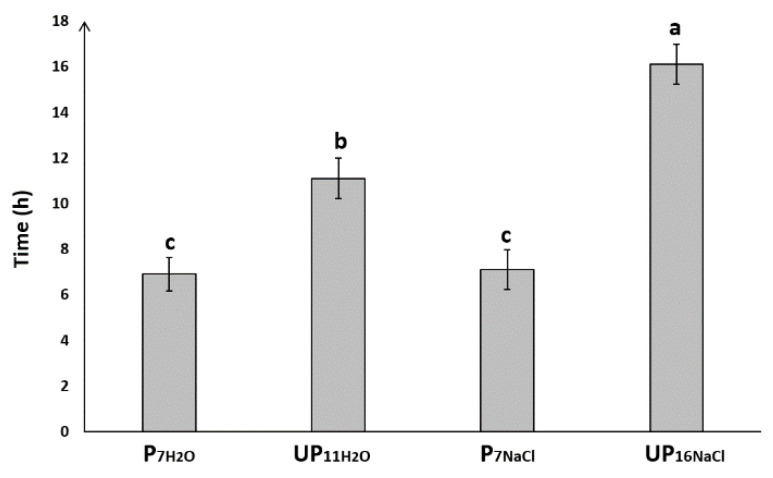
Time to reach 1% germination, (primed seeds germinating 7 h in water, P_7H___2_O_; unprimed seeds germinating 11 h in water, UP_11H___2_O_; primed seeds germinating 7 h in NaCl, P_7NaCl_; unprimed seeds germinating 16 h in NaCl, UP_16NaCl_). The differences were statistically significant, as determined by one-way analysis of variance (ANOVA) and Tukey–Kramer multiple comparison test (*n* = 10, *p* < 0.05). The vertical bars denote SD. The same letters on bars describe not significant differences between means.

**Figure 2 ijms-23-00349-f002:**
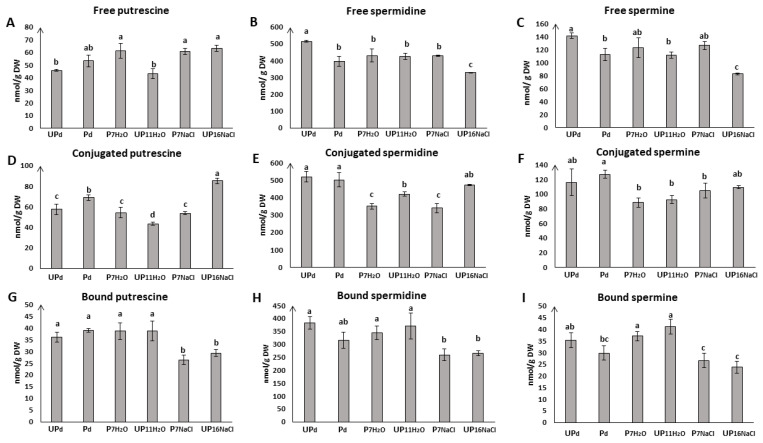
Concentration of (**A**) free putrescine, (**B**) free spermidine, (**C**) free spermine, (**D**) conjugated putrescine, (**E**) conjugated spermidine, (**F**) conjugated spermine, (**G**) bound putrescine, (**H**) bound spermidine, (**I**) bound spermine in the seeds of *Brassica napus* before the start of germination (dry unprimed seeds, UP_d_ and primed dried seeds, P_d_) as well as during germination (primed seeds germinating for 7 h in water, P_7H___2_O_; unprimed seeds germinating for 11 h in water, UP_11H___2_O_; primed seeds germinating for 7 h in NaCl, P_7NaCl_; unprimed seeds germinating for 16 h in NaCl, UP_16NaCl_). The differences were statistically significant, as determined by one-way analysis of variance (ANOVA) and Tukey–Kramer multiple comparison test (*n* = 6, *p* < 0.05). The vertical bars denote SD. The same letters on bars describe not significant differences between means.

**Figure 3 ijms-23-00349-f003:**
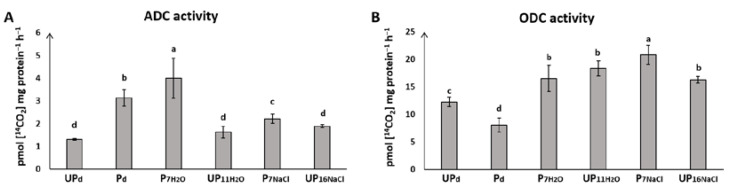
Activity of (**A**) arginine decarboxylase (ADC) and (**B**) ornithine decarboxylase (ODC) in the seeds of *Brassica napus* before the start of germination (dry unprimed seeds, UP_d_ and primed dried seeds, P_d_) as well as during germination (primed seeds germinating for 7 h in water, P_7H___2_O_; unprimed seeds germinating for 11 h in water, UP_11H___2_O_; primed seeds germinating for 7 h in NaCl, P_7NaCl_; unprimed seeds germinating for 16 h in NaCl, UP_16NaCl_). The differences were statistically significant, as determined by one-way analysis of variance (ANOVA) and Tukey–Kramer multiple comparison test (*n* = 6, *p* < 0.05). The vertical bars denote SD. The same letters on bars describe not significant differences between means.

**Figure 4 ijms-23-00349-f004:**
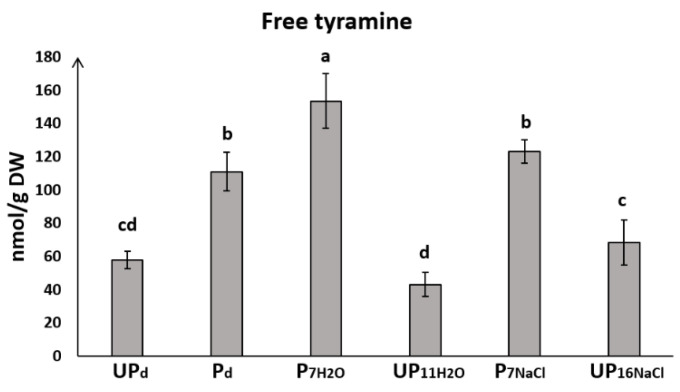
Concentration of free tyramine in the seeds of *Brassica napus* before the start of germination (dry unprimed seeds, UP_d_ and primed dried seeds, P_d_) as well as during germination (primed seeds germinating for 7 h in water, P_7H___2_O_; unprimed seeds germinating for 11 h in water, UP_11H___2_O_; primed seeds germinating for 7 h in NaCl, P_7NaCl_; unprimed seeds germinating for 16 h in NaCl, UP_16NaCl_). The differences were statistically significant, as determined by one-way analysis of variance (ANOVA) and Tukey–Kramer multiple comparison test (*n* = 6, *p* < 0.05). The vertical bars denote SD. The same letters on bars describe not significant differences between means.

**Figure 5 ijms-23-00349-f005:**
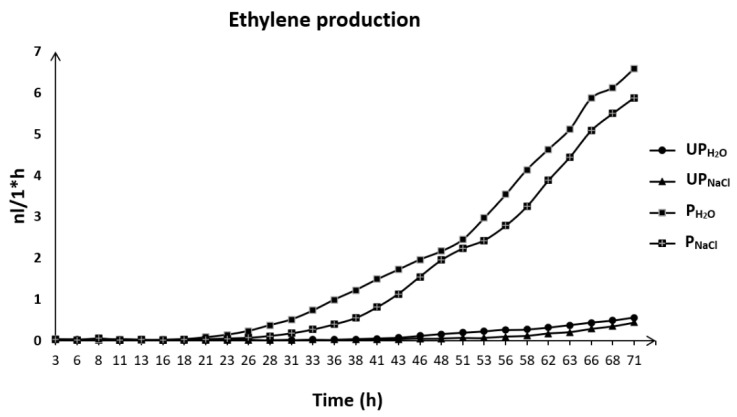
Ethylene production during germination of primed and unprimed seeds of *Brassica napus* in water and in NaCl (primed seeds germinating in water, P_H___2_O_; unprimed seeds germinating in water, UP_H___2_O_; primed seeds germinating in NaCl, P_NaCl_; unprimed seeds germinating in NaCl, UP_NaCl_). Analysis were conducted in three independent experiments (*n* = 3).

**Figure 6 ijms-23-00349-f006:**
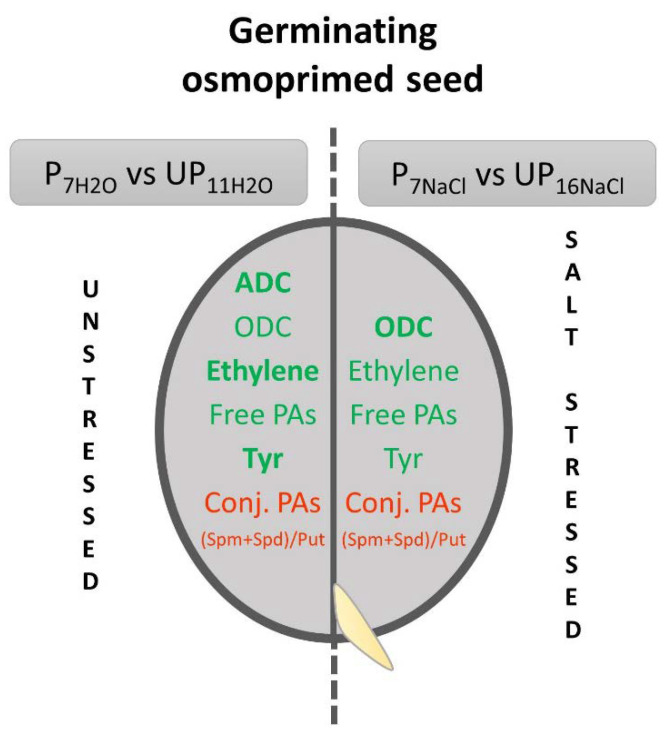
The summary of changes observed in the germinating osmoprimed *Brassica napus* seeds revealed in the present study. Stimulated/increased parameters in the germinating primed vs. unprimed variants are marked in green, whereas inhibited/decreased are marked in red. More intense changes between primed and unprimed seeds are in bold.

**Table 1 ijms-23-00349-t001:** PAs ratios calculated for seeds of *Brassica napus* before the start of germination (dry unprimed seeds, UP_d_ and primed dried seeds, P_d_) as well as during germination (primed seeds germinating 7 h in water, P_7H___2_O_; unprimed seeds germinating for 11 h in water, UP_11H___2_O_; primed seeds germinating for 7 h in NaCl, P_7NaCl_; unprimed seeds germinating for 16 h in NaCl, UP_16NaCl_). The same letters in each row of the table describe not significant differences between means.

PAs Ratio	UP_d_	P_d_	P_7H___2_O_	UP_11H___2_O_	P_7NaCl_	UP_16NaCl_
(Spd + Spm)/Put	12.266 (a)	9.215 (c)	8.912 (c)	11.676 (b)	9.102 (c)	7.218 (d)
Put/Spd	0.098 (c)	0.133 (b)	0.137 (b)	0.103 (c)	0.137 (b)	0.167 (a)

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
