# Peer review of "Endogenous Polyamines and Ethylene Biosynthesis in Relation to Germination of Osmoprimed Brassica napus Seeds under Salt Stress"

_ijms, 2021, doi:10.3390/ijms23010349_

Round 1
Reviewer 1 Report
The manuscript by Katarzyna Lechowska et al. "Endogenous Polyamines and Ethylene Biosynthesis in Relation to Germination of Osmoprimed Brassica napus Seeds under Salt Stress" is devoted to assessing the role of low molecular weight effectors under the influence of salt stress. In addition, the role of these substances for seeds subjected to osmopriming was assessed. The work was carried out at a high methodological level, this study will be useful to researchers engaged in the field of science-intensive agriculture and plant science.
-Line 35-39
Despite the fact that the authors refer to fairly authoritative sources, I would recommend placing a link to the monitoring reports of international organizations. Perhaps FAO UN generates such reports.
-The text does not disclose the advantages of osmopriming in comparison with other methods, or the reasons for choosing this method (besides the fact that it was used earlier) by the team of authors.
-Line 80-82
Must be accompanied by a link. Systematic proximity does not indicate similarities in physiological responses to stressors. Earlier you wrote that even different varieties can respond differently to osmotic stress, but here representatives of different genera are compared.
-Line 83
There is a lack of logical transition between these sections of the introduction.
- The aim of the study is the following "PAs metabolism may affect the metabolism of ethylene". Those. the pathetic nature of the influence of polyamides is implied. However, the title includes “Endogenous Polyamines and Ethylene Biosynthesis” as quite related categories. Thus, it is necessary either to correct the name or to adjust the goal.
- The quality of the drawings is not very high.
- Picture 1
The text in brackets refers to the description of the graph rather than the legend. This fragment can be included in the main text of the manuscript.
- I believe that some clarification is needed in the text on account of the difference between free, conjugated, and bound forms of polyamide.
- The text contains a fairly large number of abbreviations, I would recommend that the authors enter the appropriate section at the end of the manuscript.
- A very detailed discussion. However, it could be supplemented by a metabolic scheme demonstrating the relationship between the studied low-molecular substances/components of the stress response of rapeseed plants.
Author Response
The manuscript by Katarzyna Lechowska et al. "Endogenous Polyamines and Ethylene Biosynthesis in Relation to Germination of Osmoprimed Brassica napus Seeds under Salt Stress" is devoted to assessing the role of low molecular weight effectors under the influence of salt stress. In addition, the role of these substances for seeds subjected to osmopriming was assessed. The work was carried out at a high methodological level, this study will be useful to researchers engaged in the field of science-intensive agriculture and plant science.
Our reply: We thank the Reviewer for these comments.
Line 35-39Despite the fact that the authors refer to fairly authoritative sources, I would recommend placing a link to the monitoring reports of international organizations. Perhaps FAO UN generates such reports.
Our reply: We have added the appropriate link.
The text does not disclose the advantages of osmopriming in comparison with other methods, or the reasons for choosing this method (besides the fact that it was used earlier) by the team of authors.
Our reply: The choice of the priming treatment (250C, seed soaking in PEG solution) was based on preliminary studies on rape (Brassica napus L. cv Libomir) seeds. Seeds were primed in PEG 6000 solution (osmotic potential −1.2 MPa) during 7 days at 25◦C in the darkness on Petri dishes lined with 3 layers of filter paper wetted with PEG. During this treatment seeds were soaked in PEG solution to 2/3 of their height but a whole seed was kept wet. This priming protocol was the most effective in terms of improved germination. We have added the appropriate explanation in the Material and methods section.Moreover, in the Introduction section we have added additional information: “Priming procedure itself may generate a moderate abiotic stress during both soaking (e.g. osmotic stress and drought created by the priming agents) and dehydration steps. It is tempting to propose that priming cannot simply be considered as an acceleration of germination-related processes but also involves other specific mechanisms that improve germination and allow the seeds to cope with environmental stresses during seedling establishment.”
Line 80-82Must be accompanied by a link. Systematic proximity does not indicate similarities in physiological responses to stressors. Earlier you wrote that even different varieties can respond differently to osmotic stress, but here representatives of different genera are compared.
Our reply: This sentence does not provide significant information on the subject of the paper since this work focuses on physiological responses to salinity stress. We have removed this sentence.
Line 83There is a lack of logical transition between these sections of the introduction.
Our reply: We have added sentences connecting these two sections of the introduction.
The aim of the study is the following "PAs metabolism may affect the metabolism of ethylene". Those. the pathetic nature of the influence of polyamides is implied. However, the title includes “Endogenous Polyamines and Ethylene Biosynthesis” as quite related categories. Thus, it is necessary either to correct the name or to adjust the goal.
Our reply: The authors agree with the Reviewer and we have adjusted the aim of the study. The sentence: “As the changes in PAs metabolism may affect the metabolism of ethylene, ethylene production was also analyzed” was replaced with the sentence: “As the changes in PAs metabolism may affect the production of ethylene, ethylene emission was also analyzed.”
The quality of the drawings is not very high.
Our reply: We have improved the quality and the resolution of the drawings.
Picture 1The text in brackets refers to the description of the graph rather than the legend. This fragment can be included in the main text of the manuscript.
Our reply: To our opinion, captions should be brief, providing comprehensive explanation of the data as it appears within the text.
I believe that some clarification is needed in the text on account of the difference between free, conjugated, and bound forms of polyamide.
Our reply: We have added the requested information in the Introduction section.
The text contains a fairly large number of abbreviations, I would recommend that the authors enter the appropriate section at the end of the manuscript.
Our reply: According to the Instructions for Authors, aabbreviations have been defined the first time they appear in each of three sections: the abstract; the main text; the first figure or table. When defined for the first time, the abbreviation has been added in parentheses after the written-out form. Moreover, we have added the list of abbreviations.
A very detailed discussion. However, it could be supplemented by a metabolic scheme demonstrating the relationship between the studied low-molecular substances/components of the stress response of rapeseed plants.
Our reply: As suggested by the Reviver the paper has been supplemented by the Conclusions section with a metabolic scheme demonstrating the relationship between the studied low-molecular compounds in relation to germination of osmoprimed Brassica napus seeds under salt stress.

Reviewer 2 Report
Dear Authors
Present manuscript entitled "Endogenous Polyamines and Ethylene Biosynthesis in Relation to Germination of Osmoprimed Brassica napus Seeds under Salt Stress" demonstrated that arginine decarboxylase
pathway seems to be responsible for the accumulation of PAs in primed seeds. The study was very well planned and experiments were performed well. The presentation of results are impressive, although there are some queries, please find them below.
Figure 2C- free spermidine graph is in duplicate? or it is by mistake, as in legends its saying free spermine?
There is no information regarding how many seeds have been used for each treatment?
Number of biological/technical replicates should be mentioned in materials and methods.
Line 480- priming method should be more detailed, providing references is not enough.
Line 531, 4.4. Enzyme assay- methods should be in detail.
Short and concise conclusion may be included to convey the key findings. This may facilitate the understanding for readers.
Thank you
Author Response
Present manuscript entitled "Endogenous Polyamines and Ethylene Biosynthesis in Relation to Germination of Osmoprimed Brassica napus Seeds under Salt Stress" demonstrated that arginine decarboxylase pathway seems to be responsible for the accumulation of PAs in primed seeds. The study was very well planned and experiments were performed well. The presentation of results are impressive, although there are some queries, please find them below.
Our reply: We thank the Reviewer for these comments
Figure 2C- free spermidine graph is in duplicate? or it is by mistake, as in legends its saying free spermine?
Our reply: The description of this graph has been corrected as it refers to spermine.
There is no information regarding how many seeds have been used for each treatment? Number of biological/technical replicates should be mentioned in materials and methods.
Our reply: We have added the appropriate explanation in the Material and methods section as well as in figure captions.
Line 480- priming method should be more detailed, providing references is not enough.
Our reply: The priming method has been described in more details in Material and methods section.
Line 531, 4.4. Enzyme assay- methods should be in detail.
Our reply: We have added the description of enzyme assay methods.
Short and concise conclusion may be included to convey the key findings. This may facilitate the understanding for readers.
Our reply: As suggested by the Reviver the paper has been supplemented by the Conclusions section with a metabolic scheme demonstrating the relationship between the studied low-molecular compounds in relation to germination of osmoprimed Brassica napus seeds under salt stress.

Reviewer 3 Report
This paper reports the concentrations of free, conjugated and bound PAs in primed and nonprimed seeds of rape and intends to connect these, albeit significant but never notable (<2-fold) changes with the ethylene production and seeds germination. The causal connection is not proved by the experimental data, no clear mechanism was shown, so the conclusions are totally unfounded. I will leave alone multiple inaccuracies, like in address "Iinstitute of Biochgemistry and Biohysics" and in Fig 2c (spermidine instead of spermine), because it is not principle. The paper just failed conceptually and brings little if any useful information.
Author Response
This paper reports the concentrations of free, conjugated and bound PAs in primed and nonprimed seeds of rape and intends to connect these, albeit significant but never notable (<2-fold) changes with the ethylene production and seeds germination. The causal connection is not proved by the experimental data, no clear mechanism was shown, so the conclusions are totally unfounded. I will leave alone multiple inaccuracies, like in address "Iinstitute of Biochgemistry and Biohysics" and in Fig 2c (spermidine instead of spermine), because it is not principle. The paper just failed conceptually and brings little if any useful information.
Our reply: The present results extend our previous papers related to osmopriming improved germination of Brassica napus seeds and should be consider as a part of wider analysis that shed a new light on mechanisms underlying priming-induced improvement of seed germination. In Authors opinion the topic is original, the manuscript is well written, the methodology is well described and some interesting results are presented. The paper has been supplemented by the Conclusions section with a metabolic scheme demonstrating the relationship between the studied low-molecular compounds in relation to germination of osmoprimed Brassica napus seeds under salt stress.
